# High resolution neural connectivity from incomplete tracing data using nonnegative spline regression

**Kameron Decker Harris**
Applied Mathematics, U. of Washington
`kamdh@uw.edu`

**Stefan Mihalas**
Allen Institute for Brain Science
Applied Mathematics, U. of Washington
`stefanm@alleninstitute.org`

**Eric Shea-Brown**
Applied Mathematics, U. of Washington
Allen Institute for Brain Science
`etsb@uw.edu`

## Abstract

Whole-brain neural connectivity data are now available from viral tracing experiments, which reveal the connections between a source injection site and elsewhere in the brain. These hold the promise of revealing spatial patterns of connectivity throughout the mammalian brain. To achieve this goal, we seek to fit a weighted, nonnegative adjacency matrix among 100 μm brain "voxels" using viral tracer data. Despite a multi-year experimental effort, injections provide incomplete coverage, and the number of voxels in our data is orders of magnitude larger than the number of injections, making the problem severely underdetermined. Furthermore, projection data are missing within the injection site because local connections there are not separable from the injection signal.

We use a novel machine-learning algorithm to meet these challenges and develop a spatially explicit, voxel-scale connectivity map of the mouse visual system. Our method combines three features: a matrix completion loss for missing data, a smoothing spline penalty to regularize the problem, and (optionally) a low rank factorization. We demonstrate the consistency of our estimator using synthetic data and then apply it to newly available Allen Mouse Brain Connectivity Atlas data for the visual system. Our algorithm is significantly more predictive than current state of the art approaches which assume regions to be homogeneous. We demonstrate the efficacy of a low rank version on visual cortex data and discuss the possibility of extending this to a whole-brain connectivity matrix at the voxel scale.

## 1 Introduction

Although the study of neural connectivity is over a century old, starting with pioneering neuroscientists who identified the importance of networks for determining brain function, most knowledge of anatomical neural network structure is limited to either detailed description of small subsystems [2, 9, 14, 26] or to averaged connectivity between larger regions [7, 21]. We focus our attention on *spatial, structural* connectivity at the *mesoscale*: a coarser scale than that of single neurons or cortical columns but finer than whole brain regions. Thanks to the development of new tracing techniques, image processing algorithms, and high-throughput methods, data at this resolution are now accessible in animals such as the fly [12, 19] and mouse [15, 18]. We present a novel regression technique tailored to the challenges of learning spatially refined mesoscale connectivity from neural tracing experiments. We have designed this technique with neural data in mind and will use this

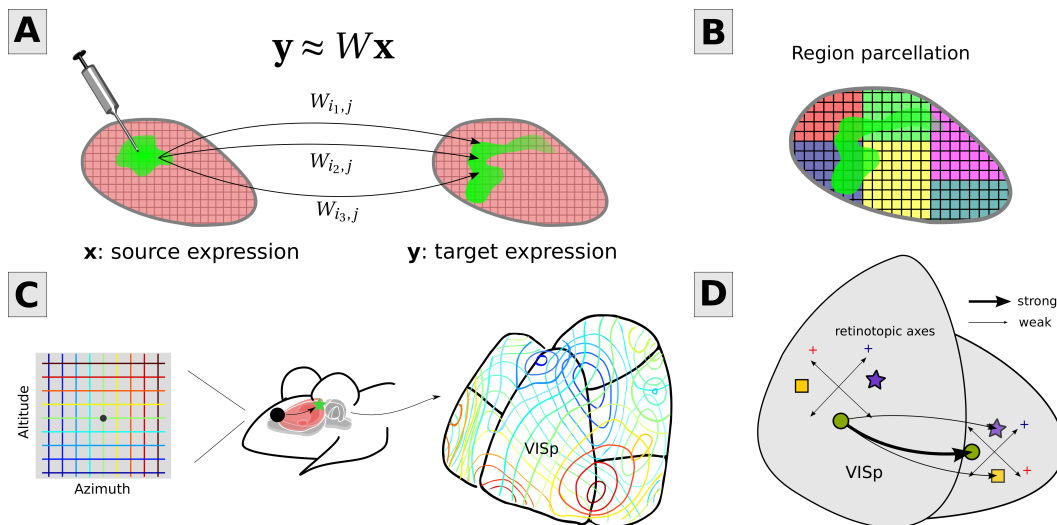

Figure 1: **A**, We seek to fit a matrix $W$ which reproduces neural tracing experiments. Each column of $W$ represents the expected signal in target voxels given an injection of one unit into a single source voxel. **B**, In the work of Oh et al. [18], a regionally homogeneous connectivity matrix was fit using a predefined regional parcellation to constrain the problem. We propose that smoothness of $W$ is a better prior. **C**, The mouse's visual field can be represented in azimuth/altitude coordinates. This representation is maintained in the retinotopy, a smoothly varying map replicated in many visual areas (e.g. [8]). **D**, Assuming locations in VISp (the primary visual area) project most strongly to positions which represent the same retinotopic coordinates in a secondary visual area, then we expect the mapping between upstream and downstream visual areas to be smooth.

language to describe our method, but it is a general technique to assimilate spatial network data or infer smooth kernels of integral equations. Obtaining a spatially-resolved mesoscale connectome will reveal detailed features of connectivity, for example unlocking cell-type specific connectivity and microcircuit organization throughout the brain [13].

In mesoscale anterograde tracing experiments, a tracer virus is first injected into the brain. This infects neurons primarily at their cell bodies and dendrites and causes them to express a fluorescent protein in their cytoplasm, including in their axons. Neurons originating in the source injection site are then imaged to reveal their axonal projections throughout the brain. Combining many experiments with different sources then reveals the pathways that connect those sources throughout the brain. This requires combining data across multiple animals, which appears justified at the mesoscale [18].

We assume there exists some underlying nonnegative, weighted adjacency matrix $W \succeq 0$ that is common across animals. Each experiment can be thought of as an injection $\mathbf{x}$, and its projections $\mathbf{y}$, so that $\mathbf{y} \approx W\mathbf{x}$ as in Fig. 1A. Uncovering the unknown $W$ from multiple experiments $(\mathbf{x}_i, \mathbf{y}_i)$ for $i = 1, \ldots, n_{\text{inj}}$ is then a multivariate regression problem: Each $\mathbf{x}_i$ is an image of the brain which represents the strength of the signal within the injection site. Likewise, every $\mathbf{y}_i$ is an image of the strength of signal elsewhere, which arises due to the axonal projections of neurons with cell bodies in the injection site. The unknown matrix $W$ is a linear operator which takes images of the brain (injections) and returns images of the brain (projections).

In a previous paper, Oh et al. [18] were able to obtain a $213 \times 213$ regional weight matrix using 469 experiments with mice (Fig. 1B). They used nonnegative least squares to find the unknown regional weights in an overdetermined regression problem. Our aim is to obtain a much higher-resolution connectivity map on the scale of voxels, and this introduces many more challenges.

First, the number of voxels in the brain is much larger than the number of injection experiments we can expect to perform; for mouse with 100 μm voxels this is $O(10^5)$ versus $O(10^3)$ [15, 18]. Also, the injections that are performed will inevitably leave gaps in their coverage of the brain. Thus specifying $W$ is underdetermined. Second, there is no way to separately image the injections and projections. In order to construct them, experimenters image the brain once by serial tomography and fluorescence

microscopy. The injection sites can be annotated by finding infected cell bodies, but there is no way to disambiguate fluorescence from the cell bodies and dendrites from that of local injections. Projection strength is thus unknown within the injection sites and the neighborhood occupied by dendrites. Third, fitting full-brain voxel-wise connectivity is challenging since the number of elements in $W$ is the square of the number of voxels in the brain. Thus we need compressed representations of $W$ as well as efficient algorithms to perform inference. The paper proceeds as follows.

In Section 2, we describe our assumption that the mesoscale connectivity $W$ is smoothly-varying in space, as could be expected from to the presence of topographic maps across much of cortex. Later, we show that using this assumption as a prior yields connectivity maps with improved cross-validation performance.

In Section 3, we present an inference algorithm designed to tackle the difficulties of underdetermination, missing data, and size of the unknown $W$. To deal with the gaps and ill-conditioning, we use smoothness as a regularization on $W$. We take an agnostic approach, similar to matrix completion [5], to the missing projection data and use a regression loss function that ignores residuals within the injection site. Finally, we present a low rank version of the estimator that will allow us to scale to large matrices.

In Section 4, we test our method on synthetic data and show that it performs well for sparse data that is consistent with the regression priors. This provides evidence that it is a consistent estimator. We demonstrate the necessity of both the matrix completion and smoothing terms for good reconstruction.

In Section 5, we then apply the spline-smoothing method to recently available Allen Institute for Brain Science (Allen Institute) connectivity data from mouse visual cortex [15, 18]. We find that our method is able to outperform current spatially uniform regional models, with significantly reduced cross-validation errors. We also find that a low rank version is able to achieve approximately $23\times$ compression of the original data, with the optimal solution very close to the full rank optimum. Our method is a superior predictor to the existing regional model for visual system data, and the success of the low rank version suggests that this approach will be able to reveal whole-brain structural connectivity at unprecedented scale.

All of our supplemental material and data processing and optimization code is available for download from: `https://github.com/kharris/high-res-connectivity-nips-2016`.

## 2    Spatial smoothness of mesoscale connectivity

The visual cortex is a collection of relatively large cortical areas in the posterior part of the mammalian brain. Visual stimuli sensed in the retina are relayed through the thalamus into primary visual cortex (VISp), which projects to higher visual areas. We know this partly due to tracing projections between these areas, but also because neurons in the early visual areas respond to visual stimuli in a localized region of the visual field called their *receptive fields* [11].

An interesting and important feature of visual cortex is the presence of topographic maps of the visual field called the *retinotopy* [6, 8, 10, 20, 25]. Each eye sees a 2-D image of the world, where two coordinates, such as azimuth and altitude, define a point in the visual field (Fig. 1C). Retinotopy refers to the fact that cells are organized in cortical space by the position of their receptive fields; nearby cells have similar receptive field positions. Furthermore, these retinotopic maps reoccur in multiple visual areas, albeit with varying orientation and magnification.

Retinotopy in other areas downstream from VISp, which do not receive many projections directly from thalamus, are likely a function of projections from VISp. It is reasonable to assume that areas which code for similar visual locations are most strongly connected. Then, because retinotopy is smoothly varying in cortical space and similar retinotopic coordinates are the most strongly connected between visual areas, the connections between those areas should be smooth in cortical space (Fig. 1C and D).

Retinotopy is a specific example of topography, which extends to other sensory systems such as auditory and somatosensory cortex [22]. For this reason, connectivity may be spatially smooth throughout the brain, at least at the mesoscale. This idea can be evaluated via the methods we introduce below: if a smooth model is more predictive of held-out data than another model, then this supports the assumption.

# 3 Nonnegative spline regression with incomplete tracing data

We consider the problem of fitting an adjacency operator $\mathcal{W} : T \times S \to \mathbb{R}_+$ to data arising from $n_{\text{inj}}$ injections into a source space $S$ which projects to a target space $T$. Here $S$ and $T$ are compact subsets of the brain, itself a compact subset of $\mathbb{R}^3$. In this mathematical setting, $S$ and $T$ could be arbitrary sets, but typically $S = T$ for the ipsilateral data we present here.[1] The source $S$ and target $T$ are discretized into $n_x$ and $n_y$ cubic voxels, respectively. The discretization of $\mathcal{W}$ is then an adjacency matrix $W \in \mathbb{R}_+^{n_y \times n_x}$. Mathematically, we define the tracing data as a set of pairs $\mathbf{x}_i \in \mathbb{R}_+^{n_x}$ and $\mathbf{y}_i \in \mathbb{R}_+^{n_y}$, the source and target tracer signals at each voxel for experiments $i = 1, \ldots, n_{\text{inj}}$. We would like to fit a linear model, a matrix $W$ such that $\mathbf{y}_i \approx W \mathbf{x}_i$. We assume an observation model

$$\mathbf{y}_i = W \mathbf{x}_i + \eta_i$$

with $\eta_i \overset{\text{iid}}{\sim} \mathcal{N}(0, \sigma^2 I)$ multivariate Gaussian random variables with zero mean and covariance matrix $\sigma^2 I \in \mathbb{R}^{n_y \times n_y}$. The true data are not entirely linear, due to saturation effects of the fluorescence signal, but the linear model provides a tractable way of "credit assignment" of individual source voxels' contributions to the target signal [18].

Finally, we assume that the target projections are unknown within the injection site. In other words, we only know $\mathbf{y}_j$ outside the support of $\mathbf{x}_j$, which we denote $\operatorname{supp} \mathbf{x}_j$, and we wish to only evaluate error for the observable voxels. Let $\Omega \in \mathbb{R}^{n_y \times n_{\text{inj}}}$, where the $j$th column $\Omega_j = 1 - \mathbb{1}_{\operatorname{supp} \mathbf{x}_j}$, the indicator of the complement of the support. We define the orthogonal projector $P_\Omega : \mathbb{R}^{n_y \times n_{\text{inj}}} \to \mathbb{R}^{n_y \times n_{\text{inj}}}$ as $P_\Omega(A) = A \circ \Omega$, the entrywise product of $A$ and $\Omega$. This operator zeros elements of $A$ which correspond to the voxels within each experiment's injection site. The operator $P_\Omega$ is similar to what is used in matrix completion [5], here in the context of regression rather than recovery.

These assumptions lead to a loss function which is the familiar $\ell^2$-loss applied to the projected residuals:

$$\frac{1}{\sigma^2 n_{\text{inj}}} \| P_\Omega(WX - Y) \|_F^2 \tag{1}$$

where $Y = \begin{bmatrix} \mathbf{y}_1, \ldots, \mathbf{y}_{n_{\text{inj}}} \end{bmatrix}$ and $X = \begin{bmatrix} \mathbf{x}_1, \ldots, \mathbf{x}_{n_{\text{inj}}} \end{bmatrix}$ are data matrices. Here $\| \cdot \|_F$ is the Frobenius norm, i.e. the $\ell^2$-norm of the matrix as a vector: $\|A\|_F = \|\operatorname{vec}(A)\|_2$, where $\operatorname{vec}(A)$ takes a matrix and converts it to a vector by stacking consecutive columns.

We next construct a regularization penalty. The matrix $W$ represents the spatial discretization of a two-point kernel $\mathcal{W}$. An important assumption for $\mathcal{W}$ is that it is spatially smooth. Function space norms of the derivatives of $\mathcal{W}$, viewed as a real-valued function on $T \times S$, are a natural way to measure the roughness of this function. For this study, we chose the squared $L^2$-norm of the Laplacian

$$\int_{T \times S} |\Delta \mathcal{W}|^2 \, \mathrm{d}y \mathrm{d}x,$$

which is called the thin plate spline bending energy [24]. In the discrete setting, this becomes the squared $\ell^2$-norm of a discrete Laplacian applied to $W$:

$$\| L \operatorname{vec}(W) \|_2^2 = \left\| L_y W + W L_x^T \right\|_F^2. \tag{2}$$

The operator $L : \mathbb{R}^{n_y n_x} \to \mathbb{R}^{n_y n_x}$ is the discrete Laplacian operator or second finite difference matrix on $T \times S$. The equality in Eqn. (2) results from the fact that the Laplacian on the product space $T \times S$ can be decomposed as $L = L_x \otimes I_{n_y} + I_{n_x} \otimes L_y$ [17]. Using the well-known Kronecker product identity for linear matrix equations

$$\left( B^T \otimes A \right) \operatorname{vec}(X) = \operatorname{vec}(Y) \iff AXB = Y \tag{3}$$

gives the result in Eqn. (2) [23], which allows us to efficiently evaluate the Laplacian action. As for boundary conditions, we do not want to impose any particular values at the boundary, so we choose the finite difference matrix corresponding to a homogeneous Neumann (zero derivative) boundary condition. [2]

Combining the loss and penalty terms, Eqn. (1) and (2), gives a convex optimization problem for inferring the connectivity:

$$W^* = \arg\min_{W \succeq 0} \|P_\Omega(WX - Y)\|_F^2 + \lambda \frac{n_{\text{inj}}}{n_x} \|L_y W + W L_x^T\|_F^2 . \tag{P1}$$

In the final form, we absorb the noise variance $\sigma^2$ into the regularization hyperparameter $\lambda$ and rescale the penalty so that it has the same dependence on the problem size $n_x, n_y$, and $n_{\text{inj}}$ as the loss. We solve the optimization (P1) using the L-BFGS-B projected quasi-Newton method, implemented in C++ [3, 4]. The gradient is efficiently computed using matrix algebra.

Note that (P1) is a type of nonnegative least squares problem, since we can use Eqn. (3) to convert it into

$$w^* = \arg\min_{w \succeq 0} \|Aw - y\|_2^2 + \lambda \frac{n_{\text{inj}}}{n_x} \|L w\|_2^2 ,$$

where $A = \text{diag}\left(\text{vec}(\Omega)\right)\left(X^T \otimes I_{n_y}\right)$, $y = \text{diag}\left(\text{vec}(\Omega)\right)\text{vec}(Y)$, and $w = \text{vec}(W)$. Furthermore, without the nonnegativity constraint the estimator is linear and has an explicit solution. However, the design matrix $A$ will have dimension $(n_y n_{\text{inj}}) \times (n_y n_x)$, with $O(n_y{}^3 n_{\text{inj}})$ entries if $n_x = O(n_y)$. The dimensionality of the problem prevents us from working directly in the tensor product space. And since the model is a structured matrix regression problem [1], the usual representer theorems [24], which reduce the dimensionality of the estimator to effectively the number of data points, do not immediately apply. However, we hope to elucidate the connection to reproducing kernel Hilbert spaces in future work.

## 3.1 Low rank version

The largest object in our problem is the unknown connectivity $W$, since in the underconstrained setting $n_{\text{inj}} \ll n_x, n_y$. In order to improve the scaling of our problem with the number of voxels, we reformulate it with a compressed version of $W$:

$$(U^*, V^*) = \arg\min_{U,V \succeq 0} \|P_\Omega(UV^T X - Y)\|_F^2 + \lambda \frac{n_{\text{inj}}}{n_x} \|L_y UV^T + UV^T L_x^T\|_F^2 . \tag{P2}$$

Here, $U \in \mathbb{R}_+^{n_y \times r}$ and $V \in \mathbb{R}_+^{n_x \times r}$ for some fixed rank $r$, so that the optimal connectivity $W^* = U^* V^{*T}$ is given in low rank, factored form. Note that we use nonnegative factors rather than constrain $UV^T \succeq 0$, since this is a nonlinear constraint.

This has the advantage of automatically computing a nonnegative matrix factorization (NMF) of $W$. The NMF is of separate scientific interest, to be pursued in future work, since it decomposes the connectivity into a relatively small number of projection patterns, which has interpretations as a clustering of the connectivity itself.

In going from the full rank problem (P1) to the low rank version (P2), we lose convexity. So the usual optimization methods are not guaranteed to find a global optimimum, and the clustering just mentioned is not unique. However, we have also reduced the size of the unknowns to the potentially much smaller matrices $U$ and $V$, if $r \ll n_y, n_x$. If $n_x = O(n_y)$, we have only $O(n_y r)$ unknowns instead of $O(n_y{}^2)$. Evaluating the penalty term still requires computation of $n_y n_x$ terms, but this can be performed without storing them in memory.

We use a simple projected gradient method with Nesterov acceleration in Matlab to find a local optimum for (P2) [3], and will present and compare these results to the solution of (P1) below. As before, computing the gradients is efficient using matrix algebra. This method has been used before for NMF [16].

## 4 Test problem

We next apply our algorithms to a test problem consisting of a one-dimensional "brain," where the source and target space $S = T = [0, 1]$. The true connectivity kernel corresponds to a Gaussian profile about the diagonal plus a bump:

$$W_{\text{true}}(x, y) = \exp\left\{-\left(\frac{x - y}{0.4}\right)^2\right\} + 0.9 \exp\left\{-\frac{(x - 0.8)^2 + (y - 0.1)^2}{(0.2)^2}\right\} .$$

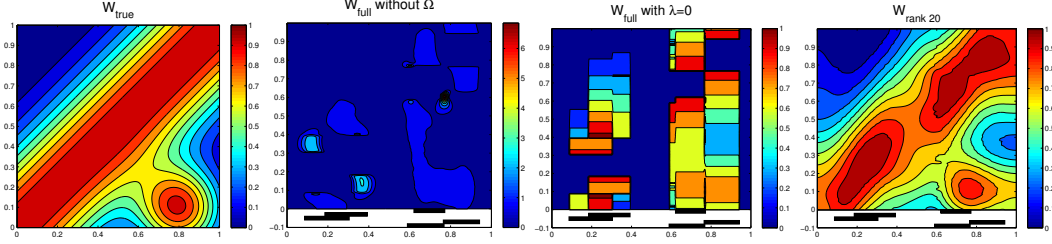

Figure 2: Comparison of the true (**far left**) and inferred connectivity from 5 injections. Unless noted, $\lambda = 100$. **Second from left**, we show the what happens when we solve (P1) without the matrix completion term $P_\Omega$. The holes in the projection data cause patchy and incorrect output. Note the colorbar range is $6\times$ that in the other cases. **Second from right** is the result with $P_\Omega$ but without regularization, solving (P1) for $\lambda = 0$. There, the solution does not interpolate between injections. **Far right** is a rank $r = 20$ result using (P2), which captures the diagonal band and off-diagonal bump that make up $W_{\text{true}}$. In this case, the low rank result has less relative error (9.6%) than the full rank result (11.1%, not shown).

See the left panel of Fig. 2. The input and output spaces were discretized using $n_x = n_y = 200$ points. Injections are delivered at random locations within $S$, with a width of $0.12 + 0.1\epsilon$ where $\epsilon \sim \text{Uniform}(0, 1)$. The values of $\mathbf{x}$ are set to 1 within the injection region and 0 elsewhere, $\mathbf{y}$ is set to 0 within the injection region, and we take noise level $\sigma = 0.1$. The matrices $L_x = L_y$ are the 5-point finite difference Laplacians for the rectangular lattice.

Example output of (P1) and (P2) is given for 5 injections in Fig. 2. Unless stated otherwise, $\lambda = 100$. The injections, depicted as black bars in the bottom of each sub-figure, do not cover the whole space $S$ but do provide good coverage of the bump, otherwise there is no information about that feature. We depict the result of the full rank algorithm (P1) without the matrix completion term $P_\Omega$, the result including $P_\Omega$ but without smoothing ($\lambda = 0$), and the result of (P2) with rank $r = 20$. The full rank solution is not shown, but is similar to the low rank one.

Figure 2 shows the necessity of each term within the algorithm. Leaving out the matrix completion $P_\Omega$ leads to dramatically biased output since the algorithm uses incorrect values $\mathbf{y}_{\text{supp}(\mathbf{x})} = 0$. If we include $P_\Omega$ but neglect the smoothing term by setting $\lambda = 0$, we also get incorrect output: without smoothing, the algorithm cannot fill in the injection site holes nor can it interpolate between injections. However, the low rank result accurately approximates the true connectivity $W_{true}$, including the diagonal profile and bump, achieving 9.6% relative error measured as $\|W^* - W_{\text{true}}\|_F / \|W_{\text{true}}\|_F$. The full rank version is similar, but in fact has slightly higher 11.1% relative error.

## 5 Finding a voxel-scale connectivity map for mouse cortex

We next apply our method to the latest data from the Allen Institute Mouse Brain Connectivity Atlas, obtained with the API at `http://connectivity.brain-map.org`. Briefly, in each experiment mice were injected with adeno-associated virus expressing a fluorescent protein. The virus infects neurons in the injection site, causing them to produce the protein, which is transported throughout the axonal and dendritic processes. The mouse brains for each experiment were then sliced, imaged, and aligned onto the common coordinates in the Allen Reference Atlas version 3 [15, 18]. These coordinates divide the brain volume into 100 μm × 100 μm × 100 μm voxels, with approximately $5 \times 10^5$ voxels in the whole brain. The fluorescent pixels in each aligned image were segmented from the background, and we use the fraction of segmented versus total pixels in a voxel to build the vectors $\mathbf{x}$ and $\mathbf{y}$. Since cortical dendrites project locally, the signal outside the injection site is mostly axonal, and so the method reveals anterograde axonal projections from the injection site.

From this dataset, we selected 28 experiments which have 95% of their injection volumes contained within the visual cortex (atlas regions VISal, VISam, VISl, VISp, VISpl, VISpm, VISli, VISpor, VISrl, and VISa) and injection volume less than 0.7 mm$^3$. For this study, we present only the results for ipsilateral connectivity, where $S = T$ and $n_x = n_y = 7497$. To compute the smoothing penalty, we used the 7-point finite-difference Laplacian on the cubic voxel lattice.

| Model | Voxel $\text{MSE}_{\text{rel}}$ | Regional $\text{MSE}_{\text{rel}}$ |
|---|---|---|
| Regional | 107% (70%) | 48% (6.8%) |
| Voxel | **33% (10%)** | **16% (2.3%)** |

Table 1: Model performance on Allen Institute Mouse Brain Connectivity Atlas data. Cross-validation errors of the voxel model (P1) and regionally homogeneous models are shown, with training errors in parentheses. The errors are computed in both voxel space and regional space, using the relative mean squared error $\text{MSE}_{\text{rel}}$, Eqn. (4). In either space, the voxel model shows reduced training and cross-validation errors relative to the regional model.

In order to evaluate the performance of the estimator, we employ nested cross-validation with 5 inner and outer folds. The full rank estimator (P1) was fit for $\lambda = 10^3, 10^4, \dots, 10^{12}$ on the training data. Using the validation data, we then selected the $\lambda_{\text{opt}}$ that minimized the mean square error relative to the average squared norm of the prediction $WX$ and truth $Y$, evaluating errors outside the injection sites:

$$\text{MSE}_{\text{rel}} = \frac{2\|P_\Omega(WX - Y)\|_F^2}{\|P_\Omega(WX)\|_F^2 + \|P_\Omega(Y)\|_F^2}. \tag{4}$$

This choice of normalization prevents experiments with small $\|Y\|$ from dominating the error. This error metric as well as the $\ell^2$-loss adopted in Eqn. (P1) both more heavily weight the experiments with larger signal. After selection of $\lambda_{\text{opt}}$, the model was refit to the combined training and validation data. In our dataset, $\lambda_{\text{opt}} = 10^5$ was selected for all outer folds. The final errors were computed with the test datasets in each outer fold. For comparison, we also fit a regional model within the cross-validation framework, using nonnegative least squares. To do this, similar to the study by Oh et al. [18], we constrained the connectivity $W_{kl} = W_{R_i R_j}$ to be constant for all voxels $k$ in region $R_i$ and $l$ in region $R_j$.

The results are shown in Table 1. Errors were computed according to both voxels and regions. For the latter, we integrated the residual over voxels within the regions before computing the error. The voxel model is more predictive of held-out data than the regional model, reducing the voxel and regional $\text{MSE}_{\text{rel}}$ by 69% and 67%, respectively. The regional model is designed for inter-region connectivity. To allow an easier comparison with the voxel model, we here include within region connections. We find that the regional model is a poor predictor of voxel scale projections, with over 100% relative voxel error, but it performs okay at the regional scale. The training errors, which reflect goodness of fit, were also reduced significantly with the voxel model. We conclude that the more flexible voxel model is a better estimator for these Allen Institute data, since it improves both the fits to training data as well as cross-validation skill.

The inferred visual connectivity also exhibits a number of features that we expect. There are strong local projections (similar to the diagonal in the test problem, Fig. 2) along with spatially organized projections to higher visual areas. See Fig. 3, which shows example projections from source voxels within VISp. These are just two of 7497 voxels in the full matrix, and we depict only a 2-D projection of 3-D images. The connectivity exhibits strong local projections, which must be filled in by the smoothing since within the injection sites the projection data are unknown; it is surprising how well the algorithm does at capturing short-range connectivity that is translationally invariant. There are also long-range bumps in the higher visual areas, medial and lateral, which move with the source voxel. This is a result of retinotopic maps between VISp and downstream areas. The supplementary material presents a view of this high-dimensional matrix in movie form, allowing one to see the varying projections as the seed voxel moves. *We encourage the reader to view the supplemental movies, where movement of bumps in downstream regions hints at the underlying retinotopy:* `https://github.com/kharris/high-res-connectivity-nips-2016`.

## 5.1 Low rank inference successfully approximates full rank solution for visual system

We next use these visual system data, for which the full rank solution was computed, to test whether the low rank approximation can be applied. This is an important stepping stone to an eventual inference of spatial connectivity for the full brain.

First, we note that the singular value spectrum of the fitted $W_{\text{full}}^*$ (now using all 28 injections and $\lambda = 10^5$) is heavily skewed: 95% of the energy can be captured with 21 of 7497 components, and 99% with 67 components. However, this does not directly imply that a nonnegative factorization will

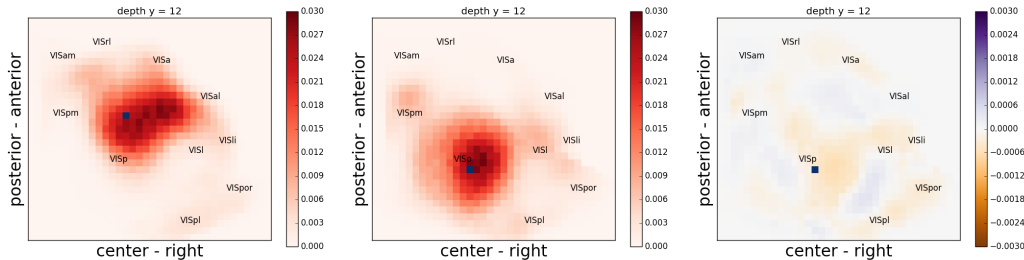

Figure 3: Inferred connectivity using all 28 selected injections from visual system data. **Left**, Projections from a source voxel (blue) located in VISp to all other voxels in the visual areas. The view is integrated over the superior-inferior axis. The connectivity shows strong local connections and weaker connections to higher areas, in particular VISam, VISal, and VISl. Movies of the inferred connectivity (full, low rank, and the low rank residual) for varying source voxel are available in the supplementary material. **Center**, For a source 800 µm voxels away, the pattern of anterograde projections is similar, but the distal projection centers are shifted, as expected from retinotopy. **Right**, The residuals between the full rank and rank 160 result from solving (P2), for the same source voxel as in the center. The residuals are an order of magnitude less than typical features of the connectivity.

perform as well. To test this, we fit a low rank decomposition directly to all 28 visual injection data using (P2) with rank $r = 160$ and $\lambda = 10^5$. The output of the optimization procedure yields $U^*$ and $V^*$, and we find that the low rank output is very similar to the full result $W^*_{\text{full}}$ fit to the same data (see also Fig. 3, which visualizes the residuals):

$$\frac{\|U^* V^{*T} - W^*_{\text{full}}\|_F}{\|W^*_{\text{full}}\|_F} = 13\%.$$

This close approximation is despite the fact that the low rank solution achieves a roughly $23\times$ compression of the $7497 \times 7497$ matrix.

Assuming similar compressibility for the whole brain, where the number of voxels is $5 \times 10^5$, would mean a rank of approximately $10^4$. This is still a problem in $O(10^9)$ unknowns, but these bring the memory requirements of storing one matrix iterate in double precision from approximately 1.9 TB to 75 GB, which is within reach of commonly available large memory machines.

## 6 Conclusions

We have developed and implemented a new inference algorithm that uses modern machine learning ideas—matrix completion loss, a smoothing penalty, and low rank factorization—to assimilate sparse connectivity data into complete, spatially explicit connectivity maps. We have shown that this method can be applied to the latest Allen Institute data from multiple visual cortical areas, and that it significantly improves cross-validated predictions over the current state of the art and unveils spatial patterning of connectivity. Finally, we show that a low rank version of the algorithm produces very similar results on these data while compressing the connectivity map, potentially opening the door to the inference of whole brain connectivity from viral tracer data at the voxel scale.

**Acknowledgements**

We acknowledge the support of the UW NIH Training Grant in Big Data for Neuroscience and Genetics (KDH), Boeing Scholarship (KDH), NSF Grant DMS-1122106 and 1514743 (ESB & KDH), and a Simons Fellowship in Mathematics (ESB). We thank Liam Paninski for helpful insights at the outset of this project. We wish to thank the Allen Institute founders, Paul G. Allen and Jody Allen, for their vision, encouragement, and support. This work was facilitated though the use of advanced computational, storage, and networking infrastructure provided by the Hyak supercomputer system at the University of Washington.

## Footnotes

[1] Ipsilateral refers to connections within the same cerebral hemisphere. For contralateral (opposite hemisphere) connectivity, $S$ and $T$ are disjoint subsets of the brain corresponding to the two hemispheres.

[2] It is straightforward to avoid smoothing across region boundaries by imposing Neumann boundary conditions at the boundaries; this is an option in our code available online.

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
