[Reviews · NeurIPS 2016]

Reviewer 1

Summary

This paper presents a structured regression model for inferring neural connectivity from viral tracing data. Combining a nonnegativity constraint, a spatial smoothness regularizer, a low-rank assumption, and a mask that handles ambiguity within the injection site, the authors derive a model that can leverage observed tracer data to infer the underlying connectivity. While the individual components of this model have been explored in other structured regression problems, this paper presents a thorough and well-executed application of these ideas to an interesting problem.

Qualitative Assessment

The underlying model is a non-negative linear regression, y = Wx + \eta, where \eta is drawn from a spherical Gaussian model. The weight matrix, W, is assumed to be nonnegative and, in probabilistic terms, drawn from a spatially smooth prior. Optionally, a low-rank assumption may be incorporated into the weight model, which can dramatically improve memory efficiency for large-scale problems. While the individual components of this model (nonnegative regression, Laplacian regularized least squares, low-rank constraints) are well-studied, I think this is a nice combination and application of these techniques to a real-world, scientific problem. The presentation of the model, the synthetic examples, and the real world applications (and supplementary movies) are particularly clear. While it is certainly valid to directly construct a objective function that captures both the reconstruction error and the domain-specific constraints and inductive biases, I think a probabilistic perspective could elucidate a number of potential extensions and connections to existing work. For example, Jonas and Kording [1] tackle a similar connectomics problem by formulating a probabilistic generative model in terms of latent variables of each neuron. While the setup is somewhat different (theirs is an unsupervised modeling problem whereas yours is framed as a regression problem), one could imagine a similar set of latent variables serving as a prior for your weight matrix. Similarly, from a probabilistic perspective, the Gaussian noise model is not particularly realistic given that the fluorescence observations are nonnegative. Likewise, the spherical covariance seems likely to be a strong assumption. While these may justifiable and computationally advantageous simplifications, it is worth discussing. Finally, the orthogonal projector, P_{\Omega}, seems like a rather fancy way of saying that your loss function is only summing over observable voxels. Minor comments: - The labels in Figure 3 are not very legible. [1] Jonas, Eric, and Konrad Kording. "Automatic discovery of cell types and microcircuitry from neural connectomics." eLife 4 (2015): e04250.

Confidence in this Review

2-Confident (read it all; understood it all reasonably well)


Reviewer 2

Summary

This paper improves the accuracy and resolution of linear models fitted to neural projection tracing experiments, by adding a smoothness prior. Additionally, low rank approximation to the connectivity matrix enables memory savings.

Qualitative Assessment

The paper is very easy to read. Also, the suggested method seems straightforward to implement, and significantly improves performance of previous region-based method. Therefore, to the best of my knowledge, the method should be useful to practitioners, especially if the code is released after publication. %% After authors' rebuttal %% I thank the authors for the rebuttal and very nice paper. My only minor suggestion is that the authors change the title so that the technical jargon appears in the end, e.g. "High resolution neural connectivity extracted from incomplete tracing data using nonnegative spline regression"

Confidence in this Review

2-Confident (read it all; understood it all reasonably well)


Reviewer 3

Summary

The authors present a novel method for identifying structural connectivity in the brain on the basis of sparse retrogade labeling experiments. The method they use combines a matrix completion loss, a smoothing penalty that enforces spatial homogeneity in the recovered structural connectivity, and a low rank approximation of the data which promises to make the method scalable to full brain maps. The method is tested on synthetic data, and applied to publicly available data from the Allen Mouse Brain Connectivity Atlas, on which it significantly outperforms state-of-the-art approaches.

Qualitative Assessment

The authors tackle a relevant and interesting problem: how best to infer anatomical connectivity at the mesoscale from brain tracing experiments, when the number of tracing experiments that can be performed is low, the number of free parameters in the connectivity matrix is a priori high, and experiments suffer from artifacts directly caused by the injection themselves. They present an elegant method that takes into account those three difficulties by means of a well-designed optimization problem. They make an assumption of spatial smoothness for the connectivity matrix that is well justified by the retinotopic organization of the visual cortex, from which the data they use comes. They address the issue of experimental artifacts at the site of injection by using a projection operator similar to those used in matrix projection techniques. They provide an extension of their optimization procedure that allows for a low-rank representation of the connectivity matrix. This extension could be used to scale the model to larger data. The paper is well written and easy to follow. The authors carefully demonstrate the various modeling choices they made on synthetic data for which ground truth is known. They conclude by providing results on real data, for which they significantly outperform the state of the art.

Confidence in this Review

2-Confident (read it all; understood it all reasonably well)


Reviewer 4

Summary

This paper applies a regularized multivariate regression method to an interesting task of recovering the neural connectivity of the visual cortex of the mouse brain at a finer level by estimating a large spare coefficient matrix. Experiment results on real data shows the improved performance over the results at the coarser regional level. --------------------------------------------------------------------------- I adjust the score after reading the rebuttal and comments of other reviewers.

Qualitative Assessment

The main contribution of this paper is to apply regularized regression method to an interesting application from neural science, which estimates the connectivity maps of mouse brain. The author extends the work of [17] by solving the problem at a finer Voxel level which corresponds to a larger regression coefficient (5000x5000 matrix). However, from the viewpoint of modeling, it is just a smoothing penalized multivariate linear regression model for the least squares problem, which is very similar to many existing matrix/tensor completion methods, thus the novelty is rather limited. For experiments, the author didn't compare with any other methods, so we don't known whether the proposed method is good enough or not. I think more algorithms and baselines should be included in the comparison.

Confidence in this Review

2-Confident (read it all; understood it all reasonably well)


Reviewer 5

Summary

The paper describes a technique for estimating a weights matrix, corresponding to brain neural connectivity. The technique combines several machine learning tools.

Qualitative Assessment

The result may be of extreme importance to neuroscientists. This is entirely out of my area, and my review relates only to the technical contribution in terms of the novelty of the proposed algorithm.

Confidence in this Review

1-Less confident (might not have understood significant parts)


Reviewer 6

Summary

This paper concerns a method to find the connectivity map between "voxels of a template brain" using several viral injections that trace the anatomy of the involved neurons. Such viral approaches to connectomics have gained popularity in the last ~10 years. Recently, the Allen Institute released one such dataset. Here, the authors refine the connectivity inference routine of that original paper.

Qualitative Assessment

The paper suggests a reasonable approach and proposes a method to infer the connectivity. The results show that the proposed method outperforms the baseline, which assumes that connectivity is homogeneous within brain regions. In this sense, the proposed method is a relaxation of the baseline. I think this approach is valuable in revealing the major connectivity themes (similar to diffusion MRI). However, the smoothness constraint also somewhat defeats the purpose of the complicated neuroanatomy. It would be helpful to discuss the linear model: it assumes that the fluorescence grows linearly with the injection sites. In the original dataset, the composite data is obtained from many brains and a template brain is formed. This should be explained. More importantly, expressions such as "the number of voxels in the brain" should be avoided.

Confidence in this Review

2-Confident (read it all; understood it all reasonably well)